# Calcium-Sensing Receptor Polymorphisms at rs1801725 Are Associated with Increased Risk of Secondary Malignancies

**DOI:** 10.3390/jpm11070642

**Published:** 2021-07-06

**Authors:** Ky’Era V. Actkins, Heather K. Beasley, Annika B. Faucon, Lea K. Davis, Amos M. Sakwe

**Affiliations:** 1Department of Microbiology, Immunology and Physiology, School of Graduate Studies and Research, Meharry Medical College, Nashville, TN 37208, USA; kactkins17@email.mmc.edu; 2Department of Biochemistry, Cancer Biology, Neuroscience and Pharmacology, School of Graduate Studies and Research, Meharry Medical College, Nashville, TN 37208, USA; hbeasley17@email.mmc.edu (H.K.B.); lea.k.davis@vumc.org (L.K.D.); 3Vanderbilt University Medical Center, Vanderbilt Genetics Institute, Nashville, TN 37232, USA; annika.b.faucon@vanderbilt.edu; 4Department of Medicine, Division of Genetic Medicine, Vanderbilt University Medical Center, Nashville, TN 37232, USA

**Keywords:** cancer-induced hypercalcemia, calcium-sensing receptor, A986S, rs1801725, metastasis, phenome-wide association study, electronic health records

## Abstract

Dysregulation of systemic calcium homeostasis during malignancy is common in most patients with high-grade tumors. However, it remains unclear whether single nucleotide polymorphisms (SNPs) that alter the sensitivity of the calcium-sensing receptor (CaSR) to circulating calcium are associated with primary and/or secondary neoplasms at specific pathological sites in patients of European and African ancestry. Multivariable logistic regression models were used to analyze the association of *CASR* SNPs with circulating calcium, parathyroid hormone, vitamin D, and primary and secondary neoplasms. Circulating calcium is associated with an increased risk for breast, prostate, and skin cancers. In patients of European descent, the rs1801725 *CASR* SNP is associated with bone-related cancer phenotypes, deficiency of humoral immunity, and a higher risk of secondary neoplasms in the lungs and bone. Interestingly, circulating calcium levels are higher in homozygous patients for the inactivating *CASR* variant at rs1801725 (TT genotype), and this is associated with a higher risk of secondary malignancies. Our data suggest that expression of CaSR variants at rs1801725 is associated with a higher risk of developing secondary neoplastic lesions in the lungs and bone, due in part to cancer-induced hypercalcemia and/or tumor immune suppression. Screening of patients for *CASR* variants at this locus may lead to improved management of high calcium associated tumor progression.

## 1. Introduction

The calcium-sensing receptor (CaSR) plays an essential role in systemic calcium homeostasis by sensing slight increases in circulating calcium levels. Besides its role in calcium homeostasis, the CaSR plays additional roles in several tissues, including several tumor types. The receptor is a tumor suppressor in certain cancers, e.g., neuroblastoma, gastric, colon, and parathyroid cancers [1,2], a tumor promoter in other cancers such as breast, prostate, ovarian, and kidney cancers [2,3,4], and plays a significant role in chemosensitivity of breast and other neoplasms [5,6,7]. For normal systemic calcium homeostasis, activation of the receptor by high calcium alters intracellular signaling pathways that reduce both parathyroid hormone (PTH) secretion by parathyroid chief cells and renal calcium reabsorption [8,9], which together restores systemic calcium to near normal levels.

During malignancy, calcium homeostasis is progressively disrupted by the secretion of osteolytic factors such as parathyroid hormone-related protein (PTHrP) by tumor cells, which like PTH promotes osteolysis. With tumor progression, the increased circulating PTHrP stimulates the biosynthesis and secretion of receptor activator of nuclear factor κB ligand (RANKL) by osteoblasts that further activate preosteoclasts osteoclast driven osteolysis [10]. This bone resorption signaling is inhibited by osteoprotegerin (OPG), which binds to RANKL and inhibits RANK signaling and bone resorption [11]. However, the release of calcium and other growth factors from bone resorption contributes to tumor growth and a vicious osteolytic cycle that ultimately results in cancer-induced hypercalcemia (CIH) or humoral hypercalcemia of malignancy [3,12,13,14]. CIH is a benign paraneoplastic syndrome common in patients with solid tumors and is associated with high mortality rates, with a median patient survival of 1–3 months from initial diagnosis [15,16]. Interestingly, high circulating calcium levels are associated with aggressive breast tumors in premenopausal women and larger breast tumors in postmenopausal women [17,18]. Although this suggests the regulation of tumor progression and/or metastasis by high calcium, whether distinct variants of the CaSR mediate the effects of high calcium remains poorly understood.

Several variants and single nucleotide polymorphisms (SNPs) have been reported in the *CASR* gene [2,19,20]. Some of these SNPs are inactivating (loss-of-function) and are associated with hypercalcemia syndromes, while others are gain-of-function or activating variants associated with hypocalcemia [21]. Specifically, missense polymorphisms at rs1801725 (e.g., A986S) and rs1801726 (e.g., Q1011E) in exon 7 of the receptor are differentially associated with calcium in various chronic diseases [22,23,24], some cancers [25,26], and various heart diseases [27]. In several other studies, the A986S CaSR variant at rs1801725 is common among subjects of European ancestry, while the Q1011E variant at rs1801726 is common in individuals of African descent [22,25,26,28,29,30]. In these studies, the A986S CaSR variant is consistently associated with calcium, while the association of the Q1011E CaSR variant with calcium has been contradictory. However, whether the expression of polymorphic *CASR* variants at these loci, which presumably affects the sensitivity of the receptor to calcium, influences cancer diagnosis at specific pathological sites in patients of European versus African descent remains unclear.

In this study, we proposed that the expression of polymorphic *CASR* variants at rs1801725 and at rs1801726 loci are associated with circulating calcium, PTH, Vitamin D and that these factors independently influence cancer diagnosis at specific pathological sites in patients of European versus African descent. Our analyses reveal that circulating calcium, PTH, and vitamin D levels are associated with neoplasms at multiple pathological sites, including breast, prostate, and skin cancers. In cancer patients, *CASR* polymorphisms at rs1801725, but not at rs1801726, are associated with circulating calcium levels, deficiency of humoral immunity, and an increased risk of secondary neoplastic lesions in the lungs and bone. Our data suggest that the CaSR variant at rs1801725, is associated with a higher risk of developing secondary neoplastic lesions in the lungs and bone due in part to cancer-induced hypercalcemia and/or tumor immune suppression. Screening of patients for *CASR* variants at this locus may lead to improved management of high calcium associated tumor progression.

## 2. Materials and Methods

### 2.1. Patient Samples

The Vanderbilt University Medical Center (VUMC, Nashville, TN, USA) maintains a clinical research warehouse known as the Synthetic Derivative (SD), which includes a de-identified mirror image of the VUMC EHR available for research purposes [31]. The SD contains over 3 million records with data on demographics, billing codes according to the International Classification of Diseases (ICD 9th and 10th revisions), procedural codes, clinical notes, medications, and laboratory measurements. In addition, over 250,000 DNA samples are linked to these records in a biorepository (BioVU).

Inclusion criteria for this study were EHRs from individuals with at least 5 ICD codes of any type on separate days over a 3-year period, which defined a “data floor” to select a population of VUMC patients enriched for primary care. ICD9 and ICD10 codes were then mapped to hierarchical phenotype classifications (i.e., “phecodes”) using the phecode map version 1.2 and 1.2b1 [32]. Phecodes were used to identify primary cancer phenotypes at specific pathological sites (e.g., “Cancer of mouth,” “Cancer of the esophagus,” etc.) and secondary malignancies (Appendix A). Cases were identified by the presence of at least one cancer phecode (which required at least two corresponding ICD codes) and controls were those who did not have any cancer phecodes. Two ancestry-specific datasets were subsequently established (Table 1), one composed of individuals of European descent herein designated European dataset (*n* = 53,682), and another containing predominantly individuals of African descent designated African dataset (*n* = 10,777).

### 2.2. Clinical Laboratory Measurements

All clinical laboratory measurements (labs), including total calcium, ionized calcium, intact parathyrin (parathyroid hormone, PTH), and 25-hydroxy vitamin D (herein referred to as vitamin D), were extracted from the SD and cleaned using the recently published QualityLab pipeline [33]. In brief, we were restricted to lab tests for which at least 70% of the observations were measured in the same unit and required that each lab was measured on a minimum of 100 patients resulting in a minimum of 1000 numeric observations recorded. We then applied lab-specific quality control filters to remove infinite and non-numeric values, as well as observations outside of 4 standard deviations from the overall sample mean, indicative of biologically implausible values due to technical or recording errors, monogenic disorders, or extreme environmental influence. Labs were adjusted for the cubic splines of age (4-knots) and transformed by inverse normalization for analysis by linear and logistic regression models. Median labs in the European and African datasets were then stratified into pre-diagnosis and post-diagnosis values, based on the earliest recorded cancer diagnosis date.

### 2.3. Genotyping and Quality Control Pipeline

Genotyping was performed by using the Illumina MEGA^EX^ array platform [34]. PLINK v1.9 was used to filter SNPs (<0.98) and individual subjects (<0.98) with low call rates, sex discrepancies, and excessive or significantly reduced heterozygosity (Fhet > 0.2). Principal component (PC) analysis was performed using flashPCA2 was used to select individuals of European ancestry and, separately, of African ancestry [35]. Minor allele frequency was calculated for *CASR* rs1801725 (c.2956G > T, p.Ala986Ser, NM_000388.4) and rs1801726 (c.3031C > G, p.Gln1011Glu, NM_000245.4) using PLINK v1.9.

### 2.4. Statistical Analysis

We first examined the relationship between the clinical labs (circulating calcium, vitamin D, and PTH levels) and cancer status (primary and secondary cancer diagnoses) after accounting for covariates including median age across the medical record, EHR-reported race, and sex. Each lab was then treated as an independent variable in separate multivariable logistic regression models of twenty-two primary and seven secondary cancer diagnoses for a total of 116 models tested.

Next, we tested the relationship between *CASR* SNPs (rs1801725 and rs1801726) and circulating calcium, ionized calcium, vitamin D, and PTH levels after adjusting for sex and the top 10 principal components (PCs) calculated from the genetic data (to adjust for population stratification) in separate multivariable linear regression models. The analysis was performed separately in European and African datasets. A total of 16 regression models were tested. In a subsequent sensitivity analysis, clinical labs were further stratified into pre- and post-cancer diagnoses according to the lab draw date.

Finally, for each SNP, we tested both an additive (rs1801725: GG vs. GT vs. TT, rs1801726: CC vs. CG vs. GG) and a recessive (rs1801725: TT vs. (GG + GT), rs1801726: GG vs. (CC + CG)) genetic model in a logistic regression framework to evaluate the relationship between the *CASR* variants and seven secondary malignant cancers in the European dataset resulting in a total of 14 models tested. Median age across the medical record, sex, and the top 10 PCs were included as covariates in genetic analyses of lab traits and cancer diagnoses.

### 2.5. Statistical Power and Correction for Multiple Comparisons

The primary analyses in this report included a total of 148 independent statistical tests, reflecting a priori derived hypotheses. A strict Bonferroni correction of *p* < 3.38 × 10^−4^ (0.05/148) was used to determine the statistical significance of primary results from four sets of regression analyses (i.e., testing associations between primary cancer and labs, *CASR* SNPs and labs, *CASR* SNPs, and primary cancer, and *CASR* SNPs and secondary malignancies). The exploratory phenome-wide association study (PheWAS), a hypothesis generating analysis, was not included in the multiple testing correction.

## 3. Results

### 3.1. Characteristics of European and African Datasets

The sample demographics of each of our datasets are shown in Table 1. Although the datasets for individuals of European and African descent differed in sample size and average age, both sets had a similar proportion of males and females. The datasets were defined by genetic ancestry clusters mapped onto principal components derived from genetic data. We analyzed the European descent and African descent populations separately in all genetic analyses to avoid population stratification confounders. In the EHR-based data without accompanying genetic data, we did not have access to ancestry information. Previous studies have reported differences in calcium and vitamin D levels by race [34,35]. Although race is not an appropriate proxy for genetic ancestry, we stratified our EHR-based phenotypic analyses by race to avoid confounding by race.

### 3.2. Allele Frequencies for rs1801725 and rs1801726

Consistent with previous reports [25], the A986S variant of the CaSR was more frequent in the European dataset compared to the African dataset. In the European dataset, the minor allele frequency (MAF) for rs1801725 was 0.14 and for rs1801726 was 0.08. In the African dataset, the MAF for rs1801725 was 0.04, while that for rs1801726 was 0.16. The European dataset had more recessive homozygotes for rs1801725 (EUR *n* = 1005; AFR *n* = 20) while the African dataset had more recessive homozygotes for rs1801726 (AFR *n* = 309; EUR *n* = 87) [28].

We then carried out a preliminary analysis of the association between rs1801725 and the rs1801726 SNPs and disease phenotypes in these datasets by using additive and recessive PheWAS models. Based on a strict Bonferroni correction (*p* < 3.65 × 10^−5^), these exploratory analyses revealed that polymorphisms at rs1801725, but not rs1801726, were strongly associated with deficiency of humoral immunity (Appendix A). Although we also observed that these CASR SNPs were associated with several other disease phenotypes, including hypercalcemia and secondary malignancy of respiratory organs, these phenotypes did not pass the strict correction threshold (Appendix A).

### 3.3. Circulating Calcium, Vitamin D, and PTH Levels in Individuals of European and African Ancestries

In a previous study, total serum calcium was found to be higher in African Americans compared to those in European American subjects [25], but the underlying cause for this difference was not further studied. Here, we determined that both circulating total (*p* = <0.001) and ionized calcium (*p* = <0.001) levels are higher in individuals of African descent compared to individuals of European descent (Table 1). Interestingly, in individuals of African descent, serum vitamin D levels are lower (*p* = <0.001), while intact PTH levels are higher (*p* = <0.001) than those in individuals of European descent (Table 1). These differences in calciotropic hormone levels suggest that the relatively higher circulating calcium in individuals of African descent may be attributed to reduced vitamin D facilitated reabsorption of calcium in the kidneys and increased PTH mediated bone resorption.

### 3.4. Association of Circulating Calcium, Vitamin D, and PTH with Primary Cancer Phenotypes

Next, we assessed whether total serum calcium, ionized calcium, vitamin D, and PTH levels are associated with specific primary cancer types (Figure 1, Appendix A). This analysis revealed that circulating calcium, vitamin D, and PTH levels are independently but mostly inversely associated with neoplasms at several pathological sites. However, total serum calcium was positively associated with breast (OR = 1.26, 95% CI = 1.23 − 1.29, *p* = 3.83 × 10^−9^), prostate (OR = 1.18, 95% CI = 1.15 − 1.21, *p* = 1.58 × 10^−40^), and skin (OR = 1.23, 95% CI = 1.21 − 1.25, *p* = 4.79 × 10^−136^) cancers (Figure 1, Appendix A). Similarly, ionized calcium was positively associated with skin cancer (OR = 1.07, 95% CI = 1.04 − 1.11, *p* = 1.47 × 10^−5^) and liver cancer (OR = 1.19, 95% CI = 1.11 − 1.28, *p* = 1.55 × 10^−6^) (Figure 1, Appendix A), while circulating vitamin D was not only associated with breast (OR = 1.13, 95% CI = 1.10 − 1.17, *p* = 9.89 × 10^−14^) and skin cancers (OR = 1.06, 95% CI = 1.03 − 1.09, *p* = 2.17 × 10^−5^), but also thyroid cancer (OR = 1.17, 95% CI = 1.10 − 1.25, *p* = 1.92 × 10^−7^) (Figure 1, Appendix A). Meanwhile, PTH was positively associated with cancer of urinary organs (OR = 1.25, 95% CI = 1.17 − 1.33, *p* = 1.16 × 10^−11^) (Figure 1, Appendix A).

We further demonstrate that circulating total and ionized calcium are also significantly, but negatively associated with secondary malignancies of lymph nodes, respiratory organs, digestive system, liver, brain/spine, and bone (Appendix A). Serum vitamin D levels are also significantly negatively associated with secondary malignancies of respiratory organs, digestive system, and liver, while circulating PTH levels are significantly associated with secondary malignancy of lymph nodes. Together, this suggests that low circulating calcium and calciotropic hormones are associated with secondary neoplasms at diverse pathological sites.

### 3.5. Association of CASR SNPS with Circulating Calcium, Vitamin D, and PTH

Several studies have reported the association of rs1801725 and rs1801726 CASR SNPs with calcium levels [25,26,27,28,29,30]. Prior to analyses at specific pathological sites, we assessed the relationship between these SNPs and the CaSR dependent variables, as well as primary cancer and secondary malignancy at all sites as single variables. The significance threshold for this analysis was set at *p* < 3.38 × 10^−4^ representing a strict Bonferroni correction (0.05/148) from a total of 148 independent statistical tests. In our European dataset, we confirmed that polymorphisms at rs1801725, but not rs1801726, were strongly associated with total and ionized calcium. The association of rs1801725 with secondary malignancy at all sites with a *p*-value of 0.03 did not pass the strict Bonferroni correction (Table 2). This analysis also revealed that Vitamin D, PTH, and primary cancers at all sites as a variable were not associated with these CaSR SNPs (Table 2). We also found that median calcium values were significantly (*p* < 0.05) different between all rs1801725 allele carriers (GG vs. GT vs. TT) and that individuals expressing the TT genotype had the highest median calcium levels compared to those expressing the GG genotype (Figure 2a, Appendix A). Although this trend was also observed for ionized calcium, only homozygous dominant and heterozygous carriers (GG vs. GT) and homozygous dominant and homozygous recessive carriers (GG vs. TT) had significantly (*p* < 0.05) different ionized calcium levels (Figure 2b). The same analytic approach also revealed a similar trend for individuals of African descent regarding total and ionized serum calcium for carriers of polymorphisms at rs1801725 (GG vs. GT) (Figure 2c, Figure 2d, Appendix A).

We also confirmed previous reports that the T-allele at rs1801725 is associated with increased total serum calcium and, interestingly, increased ionized calcium levels in the European population (Appendix A) [25]. The biggest effect was observed with the assessment of total serum calcium levels with an OR of 1.15 (95% CI = 1.13 − 1.16, *p* = 8.40 × 10^−73^), followed by ionized calcium with an OR of 1.10 (95% CI = 1.07 − 1.14, *p* = 1.43 × 10^−9^). A similar linear regression analysis revealed that polymorphisms at rs1801726 in individuals of European descent are not associated with circulating calcium, vitamin D, or PTH levels. We next showed that CASR polymorphisms at the rs1801725 locus are associated with circulating total and ionized calcium levels prior to and after cancer diagnosis in individuals of European descent (Appendix A).

For individuals of African descent, total serum calcium was modestly (*p* < 0.05) associated with rs1801725 (OR = 1.12, 95% CI = 1.06 − 1.20, *p* = 2.34 × 10^−4^). As in the European population, the rs1801726 locus was not significantly associated with calcium, vitamin D, or PTH (Appendix A). Together, this suggests that polymorphisms at rs1801725 but not at rs1801726 are associated with circulating serum calcium in individuals of European or African descent.

### 3.6. Association of CASR SNPs with Secondary Cancer Types

Although assessment of tumor characteristics and/or progression patterns are not available in the EHRs, we hypothesized that the association of CaSR SNPs with tumors at secondary sites (secondary malignancies) is indicative of the progression of primary tumors to these sites. To test this, we first assessed whether the CaSR SNPs were associated with any secondary malignancy (i.e., cases with any secondary malignancy code). This analysis revealed that the polymorphism at rs1801725 (TT genotype), but not rs1801726, was significantly associated (*p* < 0.05) with secondary malignancies in the European dataset (Appendix A).

We next determined which secondary cancers were associated with the rs1801725 CASR SNP. Using an additive logistic regression model, we observed that the rs1801725-T allele was modestly associated with the presence of secondary malignancy of respiratory organs (OR = 1.21, 95% CI = 1.08 − 1.35, *p* = 9.61 × 10^−4^) and with secondary malignancy of bone (OR of 1.20 (95% CI = 1.06 − 1.36, *p* = 4.28 × 10^−3^) in the European dataset (Figure 3, Appendix A). Using a recessive model, we confirmed that the TT genotype at rs1801725 was more common among patients with secondary malignancy of bone (OR = 1.82, 95% CI = 1.25 − 2.65, *p* = 1.80 × 10^−3^) (Appendix A). These results, while supportive of the hypothesis, did not exceed our strict multiple correction threshold. The small sample size of TT allele carriers in the population of African ancestry for rs1801725 precluded further genotype analysis in this population (Table 1).

## 4. Discussion

In this study, we assembled larger, genotyped datasets consisting of individuals of European and African descent. We interrogated whether *CASR* polymorphisms at rs1801725 and rs1801726 are associated with circulating total and ionized calcium, vitamin D, and intact PTH levels. We also investigated whether these SNPs, circulating calcium PTH, and vitamin D are associated with primary cancers at multiple pathological sites as well as with and secondary malignancies for individuals with primary cancer diagnosis. We replicated previous studies showing that circulating calcium levels in individuals of African descent are higher than those of European descent [25]. We further demonstrate that in individuals of European descent, the *CASR* variant at rs1801725, but not rs1801726, are associated with calcium and bone-related secondary cancer phenotypes, a higher risk of developing secondary neoplastic lesions in the lungs and bone, and interestingly, deficiency of humoral immunity. Overall, our data suggest that dysregulation of calcium homeostasis via expression of potentially inactivating *CASR* variants at rs1801725 (e.g., A986S) may be an important driver for the progression of primary neoplasms at several pathological sites, including breast, prostate, and skin, as well as metastasis to lung and bone tissues. This study provides further evidence suggesting that dysregulation of calcium homeostasis is distinct in cancer patients of African and European descent. The distinction is more pronounced in patients expressing polymorphic CaSR variants at rs1801725 rather than at rs1801726. Data from this study suggest that the higher circulating calcium in individuals of African descent is presumably due to lower circulating 1,25-OH vitamin D and higher PTH levels. This is especially interesting as reduced vitamin D levels are associated with reduced reabsorption of calcium in the kidneys [36], while higher circulating PTH will lead to increased bone resorption [37] and, consequently, higher circulating calcium levels.

This study used a relatively large dataset of African American patients comprising 10,777 records from this cohort. We were not able to detect any significant associations between polymorphisms at rs1801725 and circulating calcium or cancer phenotypes in this cohort primarily due to the low minor allele frequency (MAF) at this locus. Similar to other studies, the rs1801726 SNP was not associated with any of the cancer phenotypes in either the European or the African ancestry datasets [38]. However, in individuals of African descent, we found modest associations between these *CASR* SNPs and several disease phenotypes in our exploratory PheWAS. This includes a positive association between rs1801725 and osteoporosis, whereas previous reports found no associations between osteoporosis and rs1801725 in individuals of European ancestry [39]. Finally, the lack of association between rs1801726 and circulating calcium in individuals of African ancestry may suggest a lack of effect of *CASR* variants at this locus on the activity of the receptor. Although some studies have linked the development of hypercalcemia to *CASR* variants at these loci [24], direct evidence on the activity of the A986S and the Q1011E receptor variants remains to be clearly demonstrated. Overall, the expression of CASR variants at rs1801725 in >20% of cancer patients may be linked to the reduced sensitivity of the receptor to calcium, and therefore, the potential for the development of CIH. However, whether the difference in circulating calcium levels and /or the expression of *CASR* variants at whether rs1801725 underlies the disparate aggressiveness of especially prostate and breast cancers in patients of African versus European ancestries remains to be clearly delineated.

As previously reported, the CaSR plays distinct roles in several tissues, including several tumor types. It is a tumor suppressor in certain cancers (e.g., neuroblastoma, gastric, colon, and parathyroid cancers) but a tumor promoter in other cancers such as breast, prostate, ovarian, and kidney cancers [2]. In many of these tissues, CaSR functions by activating several downstream effectors depending on the coupled G proteins [40]. This biased coupling of G proteins to the activated CaSR partly accounts for the diverse role of the receptor in several tissues. Our findings are consistent with the diverse functions of CaSR in that circulating calcium is positively associated with prostate, breast, and skin cancers in which CaSR expression stimulates tumor cell growth and motility [2,3,41] while most other malignancies, including colorectal cancer in which downregulation of CaSR stimulates tumor growth are inversely associated with calcium [42,43]. Although pending further investigation, it is also possible that the expression of distinct variants of the receptor at rs1801725 may lead to distinct CaSR signaling and/or tumor progression patterns in specific subtypes of highly heterogeneous cancers such as breast cancer.

Besides the association with circulating calcium, we also found a significant association between *CASR* rs1801725 polymorphisms and deficiency in humoral immunity in our exploratory PheWAS analysis. This finding is not only novel but also intriguing. Expression of CaSR in immune cells such as neutrophils, monocytes/macrophages, and T lymphocytes [44,45] is associated with proinflammatory cytokine secretion. Previous studies reported that CaSR regulates immune response by controlling the NLR family pyrin domain containing 3 (NLRP3) in macrophages [46] and in monocytes [47]. The NLRP3 inflammasome is an innate immune mediator responsible for active interleukin (IL)-1β biosynthesis and secretion, which is critical for tumor development and immunity [48]. This is supported by studies showing that activation of NLRP3 inflammasome/IL-1β pathway promotes head and neck squamous cell carcinoma tumorigenesis while inactivation of this pathway delayed tumor growth [49]. Conditional knockout of CaSR in mouse intestinal epithelium also resulted in several defects, including immunity that was skewed towards a pro-inflammatory response [50], consistent with the tumor suppressor role of the receptor in colorectal cancer. While these studies suggest the importance of CaSR in immune cells and innate immunity, others have demonstrated the presence of circulating autoantibodies against the receptor due to either aberrant immune responses against the parathyroid glands [51] or adverse immune events associated with immunosuppressive therapy such as immune checkpoint inhibitors [52,53]. While CaSR autoantibodies may play a limited role, it is possible that the association of CaSR mutants with deficiency in humoral immunity may at least in part be related to diminished activation of NLRP3 inflammasome/IL-1β pathway and anti-tumor immune responses due to decreased sensitivity of the mutant CaSR at rs1801725 to calcium. However, the true implication of exon 7 CaSR variants in humoral immunity and whether this is associated with tumorigenesis or tumor progression requires further investigations.

In addition to its role in calcium homeostasis, vitamin D also plays a central role in innate and adaptive immune function through the vitamin D receptor [54]. Individuals predisposed to higher circulating calcium due to vitamin D deficiency may have a weakened immune system that may further increase susceptibility to the more deadly secondary malignancies. This, coupled with increased PTH and hypercalcemia, may especially compromise high-risk populations such as those of African ancestry who are disproportionately diagnosed with more aggressive tumors and, consequently, higher cancer mortality rates [55].

### Study Limitations

In spite of the robustness of our findings and opportunities for future studies, some limitations of this study are worth noting. First, we were unable to analyze PTHrP, the major osteolytic factor during malignancy. This is because PTHrP levels are not routinely measured in a clinical setting, and the small sample size in our EHR datasets warranted their exclusion during the implementation of the QualityLab pipeline. Second, although our dataset containing individuals of African descent was well powered for rs1801725 in the additive models, it was not possible for us to examine the effects of the T-allele in the recessive model because of the low MAF within the African descent population. Third, the skewed nature of the frequency of SNPs at rs1801725 or rs1801726 in individuals of African and European ancestries made it difficult to generalize the effects of the rs1801725 *CASR* variants in the African population. Fourth, due to differing linkage disequilibrium structures in European and African populations, it is possible that the associations we observed with the European descent do not tag the same causal allele associations in the African descent dataset. Finally, the EHR-based data from which we drew our study population are a clinically ascertained sample; thus, ascertainment biases could impact this study. More functional in vitro and in vivo studies are needed to understand causal links between CaSR genetic variants and cancer progression and/or metastasis.

## 5. Conclusions

In summary, this study provides robust population-level genetic evidence supporting the notion that expression of the inactivating CaSR rs1801725 SNP predisposes breast, prostate, and skin cancer patients to secondary neoplastic lesions in the lung and bone tissues. Whether the development of secondary malignancies at these sites is exclusively mediated by cancer-induced hypercalcemia via CaSR signaling or includes other complications such as CaSR-mediated stimulation of inflammation and/or tumor immune suppression remains to be fully elucidated.

## Figures and Tables

**Figure 1 jpm-11-00642-f001:**
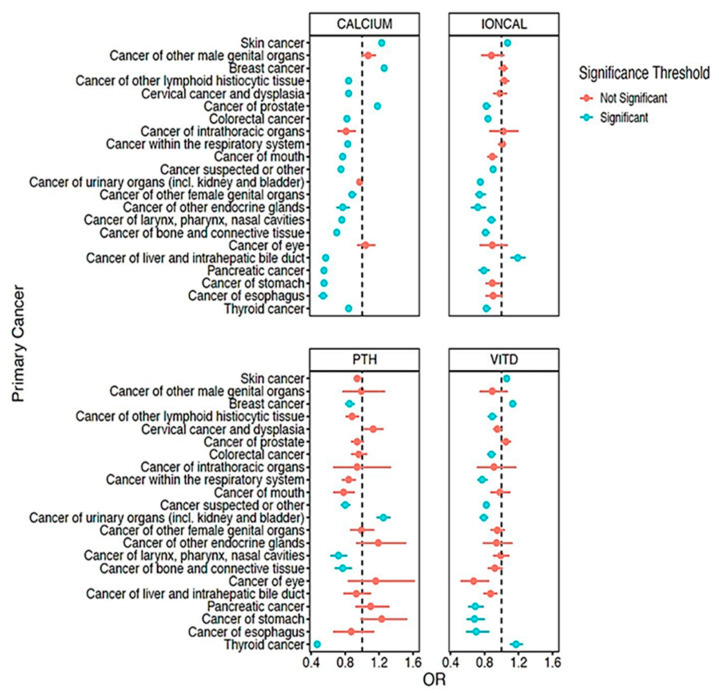
Association of circulating calcium, parathyroid hormone, and vitamin D levels with cancer risk. Forest plots showing the odds associated with circulating levels of the indicated laboratory values and diagnosis of cancer at the indicated pathologic sites. The dashed line in each plot represents an odds ratio (OR) of 1. PTH = intact parathyroid hormone, VITD = vitamin D, IONCAL = ionized calcium.

**Figure 2 jpm-11-00642-f002:**
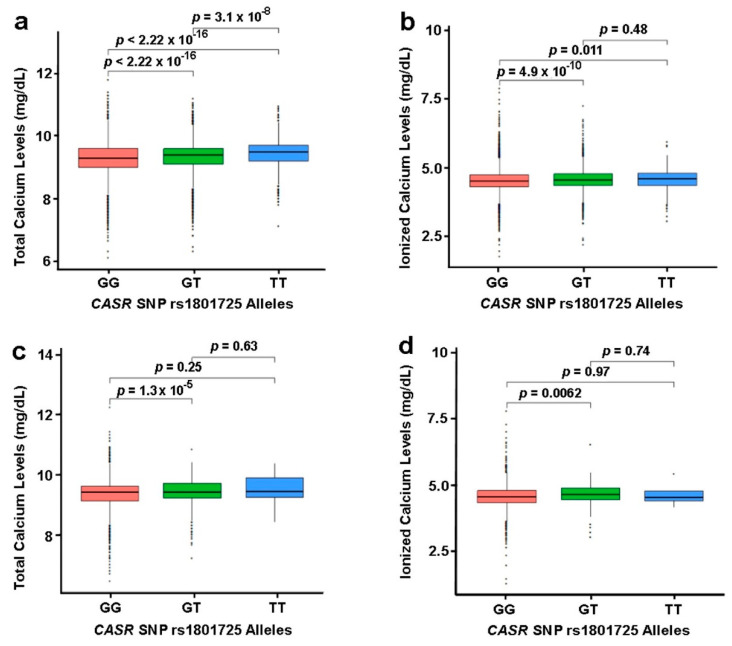
Circulating total calcium and ionized calcium in CaSR rs1801725 allele carriers. Top row (**a**,**b**): European dataset; bottom row (**c**,**d**): African dataset. The Wilcoxon rank sum test was used to compare the median circulating total (**a**) and ionized (**b**) calcium levels in individuals of European descent, and total (**c**) and ionized (**d**) calcium levels in individuals of African descent expressing the indicated polymorphisms at rs1801725 depicted by the genotype. *p* < 0.05 was considered statistically significant.

**Figure 3 jpm-11-00642-f003:**
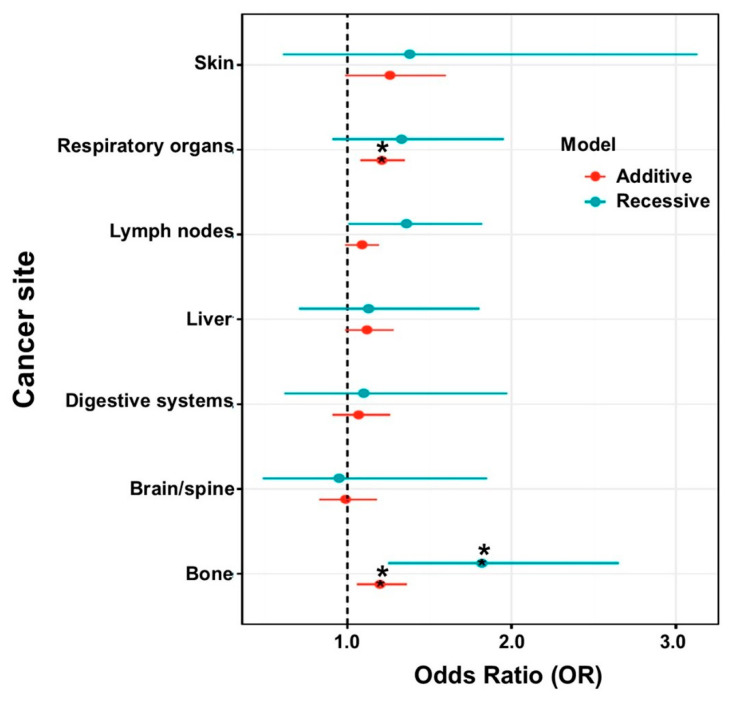
Association of circulating calcium levels with secondary cancer phenotypes. Logistic regression models were performed on secondary cancer and the CASR variants at rs1801725. The additive (red) and recessive (green) models were adjusted for median age, sex, and the first ten principal components. ***** indicates *p* < 0.01 in the logistic regression model. OR = odds ratio.

**Table 1 jpm-11-00642-t001:** Descriptive statistics for European and African descent datasets.

Demographics	European Dataset	African Dataset	*p*
N	53680	10777	
Average Age (SD)	48.74 (22.03)	37.98 (21.57)	<0.001
Average BMI (SD)	27.75 (7.25)	29.04 (8.65)	<0.001
Sex (%)			<0.001
Female	30801 (57.4)	6767 (62.8)	
Male	22879 (42.6)	4010 (37.2)	
rs1801725 Alleles (%)			<0.001
GG	39939 (74.4)	9976 (92.6)	
GT	12736 (23.7)	773 (7.2)	
TT	1005 (1.9)	20 (0.2)	
rs1801726 Alleles (%)			<0.001
CC	49733 (92.7)	7487 (69.5)	
CG	3858 (7.2)	2972 (27.6)	
GG	87 (0.2)	309 (2.9)	
EHR-Reported Race (%)			<0.001
White	52461 (97.7)	285 (2.6)	
Black	59 (0.1)	10313 (95.7)	
Calcium (mg/dL)			
N	51176	9878	
Median [IQR]	9.30 (9.05, 9.60)	9.40 (9.10, 9.60)	<0.001
Ionized Calcium (mg/dL)			
N	13899	2336	
Median [IQR]	4.52 (4.32, 4.73)	4.57 (4.35, 4.80)	<0.001
Vitamin D (ng/mL)			
N	18885	3341	
Median [IQR]	31 (24, 39)	23 (17, 30.50)	<0.001
Parathyroid Hormone (pg/mL)			
N	6937	1430	
Median [IQR]	56 (36, 91)	82 (50, 160)	<0.001

Categorical data and allele information were statistically analyzed by chi-square tests, lab data were analyzed by Wilcoxon rank sum tests, and age and BMI were analyzed by Student’s *t*-test. SD = standard deviation, EHR = electronic health record, IQR = interquartile range.

**Table 2 jpm-11-00642-t002:** Relationship between CASR exon 7 variants and calcium, PTH, vitamin D, and cancer in European descent individuals.

	rs1801725		rs1801726	
	OR (95% CI)	*p*	OR (95% CI)	*p*
Calcium	1.15 (1.13–1.16)	8.40 × 10^−73^*	0.98 (0.96–1.01)	0.16
Ionized Calcium	1.10 (1.07–1.14)	1.43 × 10^−9^*	1.03 (0.97–1.09)	0.28
Vitamin D	0.99 (0.96–1.02)	0.50	0.98 (0.94–1.03)	0.51
Parathyroid Hormone	1.00 (0.95–1.05)	0.98	1.03 (0.94–1.12)	0.57
Primary Cancer	1.02 (0.97–1.08)	0.40	1.00 (0.91–1.09)	0.95
Secondary Malignancy	1.08 (1.01–1.15)	0.03	1.00 (0.88–1.14)	0.97

The additive regression model adjusted for median age, sex, and the first ten principal components was performed between *CASR* SNPs (rs180125 (G vs. T) and rs1801726 (C vs. G)) and the indicated laboratory values as well as cancer diagnosis. In this analysis, the phecodes used for the primary cancers and secondary malignancies were each collapsed into a single diagnostic code. * *p* < 3.38 × 10^−4^ was considered statistically significant.

## Data Availability

The data that support the findings of this study are available from Vanderbilt University Medical Center, but restrictions apply to the availability of these data, which were used under license for the current study, and so are not publicly available. Data are, however, available from the authors upon reasonable request and with permission of Vanderbilt University Medical Center.

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
