# Peer review of "Calcium-Sensing Receptor Polymorphisms at rs1801725 Are Associated with Increased Risk of Secondary Malignancies"

_jpm, 2021, doi:10.3390/jpm11070642_

Round 1
Reviewer 1 Report
Please, find my comments attached.

Reviewer 2 Report
In this paper by Actkins et al, the authors describe the critical nature of calcium-sensing receptor (CaSR) polymorphisms in regulating systemic calcium homeostasis and in turn leading to an increased risk of secondary malignancies. Overall, the study is quite well rounded and clear in its analyses with the limitations listed as well.
- A minor suggestion for the authors to provide a diagram to illustrate the summary of their findings in the European and African patient datasets respectively. This will help to provide a clear idea what are the similarities and differences between the two patient populations
- Also, the Discussion section could go into a deeper literature survey hypothesizing why/how CasR mechanisms could be mechanistically leading to the different phenotypes they may observe. I think the tie-in of their database analysis to experimental findings will only add to the significance of their work here.
Reviewer 3 Report
This is an excellent paper that I really appreciated.
Basing on their previous results and on the background reporting dysregulation of systemic calcium homeostasis during malignancy in patients with high grade tumors, the Authors aimed to define whether single nucleotide polymorphisms (SNPs) that alter the sensitivity of the calcium-sensing receptor (CaSR) to circulating calcium can be associated with primary and/or secondary neoplasms at specific pathological sites in patients of different genetic ancestry , European or African.
They provide robust population-level genetic evidence supporting the notion that expression of the inactivating CaSR rs1801725 SNP predisposes breast, prostate, and skin cancer patients to secondary neoplastic lesions in the lung and bone tissues. In particular, in patients of European descent, the rs1801725 CASR SNP is associated with bone-related cancer phenotypes, deficiency of humoral immunity, and a higher risk of secondary neoplasms in the lungs and bone.
In this paper the role of vitamin D as a regulator in innate and adaptive immune function through the vitamin D receptor has been also outlined. The hypothesis that individuals predisposed to higher circulating calcium due to vitamin D deficiency may have a weakened immune system that may further increase susceptibility to secondary malignancies is in line with the evidence that increased PTH and hypercalcemia may contribute in high risk populations such as those of African ancestry who showed lower vitamin D levels and disproportionately have higher cancer mortality rates and aggressive tumors.
I have only minor suggestions
- Line 98 parathyrin: please re-edit parathyroid hormone
- Line 229 edit those not that
- Pag 8 Fig 2 please add a subtitle in the figure : top European descent , below African descent
Round 2
Reviewer 1 Report
After reading the revised form of the manuscript entitled “Calcium-Sensing Receptor Polymorphisms at rs1801725 are Associated with Increased Risk of Secondary Malignancies” and the authors' response, I found that my comments in the previous review process were addressed, and the authors responded well to the critiques. Therefore, I recommend the acceptance of the manuscript after minor revision (text editing).